# Recycling of Post-Consumer Polystyrene Packaging Waste into New Food Packaging Applications—Part 2: Co-Extruded Functional Barriers

Frank Welle

Fraunhofer Institute for Process Engineering and Packaging IVV, Giggenhauser Straße 35, 85354 Freising, Germany; frank.welle@ivv.fraunhofer.de

**Abstract:** Post-consumer polystyrene (PS) recyclates behind a functional barrier of virgin PS polymer is an attractive way to introduce post-consumer PS recyclates in packaging materials. However, until now, there has been no official guidance published on how to test the performance of a functional barrier. In addition, no threshold limits for the evaluation of post-consumer recyclates behind FBs have been published by competent authorities. This is a drawback in the food law compliance evaluation and novel technology notification of post-consumer recyclates behind a functional barrier. In this study, co-extruded yogurt cups with an artificially spiked core layer were investigated with respect to migration of the applied surrogates. The applied migration kinetic testing method into the gas phase was shown to be sensitive and suitable for the evaluation of the functional barrier performance. For consumer safety evaluation, two worst-case scenarios were used. As a result, due to the high processing temperatures used during co-extrusion, the virgin PS functional barrier layer was partly contaminated with the surrogates from the core layer. However, on the basis of the conditions, data, and the evaluation presented, the use of post-consumer recycled PS behind an FB of virgin PS can be considered as safe.

**Keywords:** high impact polystyrene; migration; functional barrier; yogurt cups; exposure evaluation





## 1. Introduction

The recycling of packaging into new packaging materials is one of the greatest challenges facing our society. The change from a linear economy to a circular economy in the food packaging area is the primary goal. The European Commission identified plastic recycling as an essential pre-requisite for the transition to a circular economy and published an action plan to increase the amount of recyclates on the market [1]. Since post-consumer recycled food packaging must not endanger the health of the consumer, recycling and/or manufacturing processes must be developed that reduce the concentrations of potential hazardous contaminants from the first use of the recyclate in the packaging material. Such processes might be referred to either as super-clean processes [2], which reduce the concentration of substances from the first use (or from misuse) of the packaging materials in the recyclates during recycling, or so-called functional barriers (FBs) [3,4], which decrease the migration of contaminants from the packaging materials into food. In case of polyethylene terephthalate (PET), several super-clean processes are available and have been evaluated as safe [5–7]. Food packaging with a functional barrier of virgin PET has been in use for a long time [8–12]. For other polymers, such as polystyrene (PS), however, suitable recycling technologies are under development but not available on the market in an industrial scale [13]. Placing post-consumer PS recyclates behind an FB is, therefore, an attractive way to introduce post-consumer PS recyclates in packaging materials. In general, an FB prevents or reduces the transfer of substances so that their concentration in food will be below the legal specific migration limits [3,4,14]. In principle, FBs might be:

- Co-extruded structures with a virgin polymer layer of the same polymer in contact to food;
- Co-extruded structures with the post-consumer recyclate behind a suitable barrier polymer layer;
- Laminated structures with post-consumer recyclates behind a suitable barrier polymer;
- Coating of the surface with barrier lacquers or inorganic structures.

FBs based on co-extruded virgin polymer layers of the same polymer will work only for low-diffusive polymers, such as PET or PS, since the low diffusivity of the polymer reduces the diffusion or permeation of potential contaminants from the recyclate core layer into food. The thickness of the FB layer should be designed in such way that the concentration of contaminants in food at the end of the recommended shelf-life are below the legal threshold limits. As the diffusion process starts immediately after manufacturing of the packaging material, the pre-storage time before the food is packed should be taken into account when evaluating migration. Another point that should be considered when evaluating co-extruded FBs is that diffusion in the polymer melt is very high during the co-extrusion of the FB structure. As a consequence, the outer virgin FB layer might be partly contaminated with substances from the recyclate core layer. This might have a significant influence on the effectiveness of the FB.

High-diffusive polymers, such as polyethylene (PE) or polypropylene (PP), have lower barrier properties than PET or PS. Therefore, a virgin polyolefin layer is not suitable as an FB. For PE and PP, special barrier polymer should be applied, e.g., polyamide (PA) or ethylene vinyl alcohol (EVOH). Typically, such PA and EVOH layers are placed between two polyolefin layers, and the recyclate is placed behind the FB so that contaminants from the recyclate must permeate through the FB layer. Layer structures can be manufactured as co-extruded layers or as laminated layers. In the case of co-extruded layers, as mentioned above, high temperatures during manufacturing results in high diffusion rates, and the FB layer might be contaminated. Laminated structures are typically manufactured at ambient temperatures, which eliminates or reduces this problem. Coatings, such as barrier lacquers or inorganic layers (e.g., SiOx, Al, and AlOx), are also typically applied on the surface of recyclate containing packaging materials at ambient temperatures. Contamination of the inorganic FB is not a problem, but the barrier performance of inorganic coatings might be adversely affected by small defects, such as pinholes. If the films or sheets are stacked or on a roll, it should be recognized that the recyclate layer has contact with the virgin layer, and, therefore, invisible set-off might occur. This can be avoided by placing the recyclate between two FB layers.

Post-consumer recyclates behind an FB was not regulated under the scope of Regulation (EC) 282/2008 [15], and, consequently, the performance of FBs had to be in compliance with the Plastics Regulation (EC) 10/2011 [16]. However, in October 2022, the new Recycling Regulation (EU) 2022/1616 [17] came into force, repealing Regulation (EC) 282/2008 and including requirements for recyclates behind an FB. Within this new regulation, FBs are considered "novel technologies", which require an authorization of the technology or application. The authorization process involves not only a notification and an evaluation of the initial safety report by the European Commission and the competent authority before the use of recycled plastic in food contact materials is allowed but also monitoring and reporting for a minimum of two years before the European Commission decides, based upon an assessment of the European Food Safety Authority (EFSA), whether the FB technology will be classified as "suitable technology" [17]. The administrative path is defined by the Regulation (EU) 2022/1616. However, in contrast, there is no guidance on how to test the performance of an FB. In addition, EFSA has not published any threshold limits for the evaluation of post-consumer recyclates behind FBs. Any first ruling of EFSA for PET or any other polymer will finally determine how to evaluate safety of functional barrier set-ups.

Since under the old Regulation (EC) 282/2008, recyclates behind FBs were regulated under the Regulation (EC) 10/2011, some guidance on permissible maximum concentration levels in food can be gleaned. For example, in Article 27 of the preamble of Regulation (EC)

10/2011, it is mentioned that "behind a functional barrier, non-authorized substances may be used, provided they fulfil certain criteria and their migration remains below a given detection limit. Taking into account foods for infants and other particularly susceptible persons, as well as the large analytical tolerance of the migration analysis, a maximum level of 0.01 mg/kg in food should be established for the migration of a non-authorized substance through a functional barrier" [16]. A target limit of 0.01 mg/kg (10 µg/kg, 10 ppb), therefore, should be a suitable end point criteria for FB testing. For the starting concentration of the substances in the packaging materials, however, no guidance is given in Regulation (EC) 10/2011. However, similar to challenge tests in mechanical recycling, a worst-case scenario should be applied [18,19].

Another procedure for the evaluation of the performance of FBs might be the EFSA approach for mechanical recycling processes. In this regard, EFSA published detailed evaluation criteria for post-consumer PET and HDPE recyclates in direct contact with food [19,20]. This approach is more general compared with a single specific migration limit of 10 µg/kg, and it is based on the following three considerations:

- Input concentration of potential contaminants in post-consumer polymers;
- Cleaning efficiencies of the (super-clean) recycling processes;
- Exposure scenario of the consumer.

However, the EFSA evaluation principles as defined above cannot be applied to FB performance testing, since typically no special (super-clean) processes are applied in the FB manufacturing processes, and, therefore, the assessment of cleaning efficiency is not applicable, i.e., there is little or no reduction of the concentration of potential contaminants through the recycling and manufacturing process of the packaging material. If the assessment of the cleaning efficiency is replaced by the assessment of the FB performance, then the EFSA exposure approach might be also suitable for the evaluation of FBs. This is, of course, a matter of conjecture since, as mentioned above, until now, FBs were not evaluated by EFSA, and their evaluation criteria, for example, regarding PS recyclates behind a functional barrier, are not known. Thus, while the lack of clear evaluation criteria makes an evaluation difficult, the availability of guidance given in Regulation (EC) 10/2011 and the EFSA evaluation criteria for mechanical recycling processes for PET and HDPE [19,20] suggested that the evaluation of post-consumer PS recyclates behind FBs seems to be possible.

Post-consumer PS yogurt cups can be considered as a promising input stream for the recycling into new yogurt cups. Thus, the aim of this study was to determine the experimental migration kinetics of co-extruded high impact PS (HIPS) yogurt cups by use of artificially spiked layer structures. The migration kinetic data were used for the risk assessment of contaminants migrating from the contaminated core layer through the FB layer in PS yogurt cups. Post-consumer PS recyclates in direct contact with food was part of the first study [12], while PS recyclates behind an FB is another part of this study.

## 2. Results

### 2.1. Study Design

In the EFSA approach, the migrated amount is calculated by use of diffusion models [19,20]. A pre-requisite of such a diffusion modelling approach is that the concentration of the migrants in the packaging materials is homogenous, or the concentration profile is known in detail [21]. In addition, the thicknesses of the layers need to be known. Within this study, a co-extruded layer of virgin PS acted as the FB. However, it is expected that during sheet extrusion and subsequent thermoforming to yogurt cups, the virgin layer will be (partly) contaminated with substances from the core layer. Furthermore, due to the different mobility of organic substances in polymers, the concentration profiles will most likely be different for possible contaminants and FB layer thickness. Determination of the concentration profiles will, therefore, be extremely difficult or even impossible. Since the use of diffusion modelling cannot be applied for FB performance evaluation of co-extruded sheets and cups, experimental determination of the amounts of contaminants migrated

from the recyclate core layer into food is, of course, possible and, thus, absolutely necessary. Another reason that diffusion modelling cannot be used is that thermoforming of the PS sheet into a yogurt cup changes the thicknesses of the layers such that they are not constant over the whole package.

The amount of substances migrated from packaging materials into food is typically measured in food simulants. Realistic storage conditions are typically simulated by enforced testing conditions with higher temperatures and shorter contact times. Regulation (EC) 10/2011 [16] specifies the framework for the time/temperature conditions for compliance evaluation. For yogurt cups, 10 days at 40 °C with 50% ethanol as food simulant is considered as the laboratory testing conditions. These testing conditions cover all storage times at refrigerated and frozen conditions, including hot-fill conditions and/or heating up to 70 °C $\leq$ T $\leq$ 100 °C for maximum t = 120/(2((T − 70)/10)) minutes, which consists of 20 min at 100 °C, 30 min at 90 °C, 60 min at 80 °C, and 120 min at 70 °C [16].

Testing into food simulants, however, has also its pitfalls. PS is very sensitive towards solvents such as ethanol. Solvent-based food simulants are well known to change the properties of certain packaging materials. In the case of PS polymers, significant swelling effects have been reported [22,23], which causes experimental migration testing on yogurt cups to be highly over-estimative. In a previous study, we conducted migration kinetics into 50% ethanol as well as other simulants, such as *iso*-octane and 95% ethanol. The weight increase of the high impact PS at 40 °C with 50% ethanol was 1.2% [23]. Due to the fact that swelling most likely occurs on the surface of the PS package, FB barriers are much more influenced by this effect, and the barrier effect of the PS virgin layer might very well be negated [22]. On the other hand, the weight increase in contact with milk and cream was determined to be 0.1% and <0.1%, respectively [23]. So, milk products do not significantly swell the polymer surface. Experimental migration testing with 50% ethanol at elevated temperatures, therefore, is not recommended for the performance testing of co-extruded PS virgin layer. Migration testing into yogurt is not an alternative due to spoiling at high temperatures of 40 °C. Real-time storage for 40 days at 8 °C also seems not to be a suitable alternative testing due to the long storage times.

In order to circumvent the above-mentioned problems in performance testing of PS co-extruded virgin polymer FBs, an alternative procedure has been used in this study. A migration method into the gas phase was applied to determine the migrated amount of contaminants from the yogurt cups. Since the gas phase contact has no swelling effect on the surface of the polymer, FB performance can be determined. Another advantage of the gas phase migration kinetics is that the method is fully automated, and every 40 min, a kinetic point is determined. This results overall in 360 kinetic points for a 10-day migration kinetic. This provides additional information on the migration curve over time, which is not available from single endpoint migration tests typically applied in compliance evaluation of packaging materials. This precise migration kinetic allows conclusions to be drawn about the concentration profiles of the artificially spiked surrogates in the thermoformed yogurt cup.

The migration was determined by use of artificially spiked surrogates similar to challenge tests applied for the evaluation of mechanical recycling processes, including decontamination [18]. This approach is also sufficiently conservative (worst-case) to accommodate the range of likely concentrations of potential contaminants in realistic input materials and PS sheets manufactured thereof. The approach also allows one to learn about the molecular mass dependency of the migration of contaminants. The artificially spiked PS sheets were subsequently used for manufacturing into yogurt cups. For these experiments, four individual sheets and corresponding yogurt cups with different FB thicknesses were manufactured. By following this procedure, the entire manufacturing process for yogurt cups was considered. These different layer thickness were investigated in order to evaluate the minimum barrier thicknesses after co-extrusion and thermoforming. The thermoformed PS cups with spiked core layers were used for migration testing for 10 days at 40 °C. In

order to obtain an impression of the influence of the temperature, migration kinetics for 10 days at 60 °C were also measured.

### 2.2. Spiking of the Yogurt Cups

Within the study, PS sheets were artificially spiked with model substances (surrogates) which are typically used for the evaluation of the cleaning efficiencies of recycling processes. The spiking was achieved during the sheet manufacturing process. Four different sheets were manufactured with different nominal FB layer thicknesses. In addition, one sample was manufactured as a reference sample without an FB of virgin PS. In this reference sample, the spiked surrogates were in direct contact with food. Subsequently, the sheets were manufactured into yogurt cups. The five yogurt cup samples with different FB nominal thicknesses of 0 µm (reference), 20 µm, 30 µm, 40 µm, and 50 µm were analyzed regarding their concentrations of the applied surrogates. The concentrations in the yogurt cup samples are provided in Table 1.

**Table 1.** Experimentally determined mean concentrations of spiked surrogates and styrene in the investigated yogurt cups.

| Sample | Spiked Mean Concentrations in the Yogurt Cups (mg/kg) | | | | | | |
| --- | --- | --- | --- | --- | --- | --- | --- |
| | Toluene | Chlorobenzene | Styrene [1] | Methyl Salicylate | Phenyl Cyclohexane | Benzophenone | Methyl Stearate |
| Sample 0 | 872 ± 11 | 935 ± 18 | 736 ± 20 | 1290 ± 19 | 1389 ± 19 | 1370 ± 10 | 1098 ± 9 |
| Sample 1 | 561 ± 6 | 588 ± 8 | 677 ± 2 | 754 ± 10 | 807 ± 8 | 805 ± 10 | 654 ± 4 |
| Sample 2 | 472 ± 12 | 503 ± 13 | 691 ± 18 | 651 ± 20 | 699 ± 17 | 693 ± 11 | 563 ± 6 |
| Sample 3 | 379 ± 3 | 406 ± 8 | 677 ± 5 | 525 ± 6 | 564 ± 8 | 554 ± 7 | 447 ± 6 |
| Sample 4 | 264 ± 10 | 282 ± 7 | 660 ± 20 | 354 ± 9 | 382 ± 8 | 369 ± 4 | 301 ± 3 |

[1] Residual monomer, not artificially added.

The results show that the PS cup samples were successfully spiked with the surrogates. The surrogate concentrations in the yogurt cup samples were between approx. 900 mg/kg and 1400 mg/kg. The highest concentrations (for each surrogate) were determined for Sample 0, which was the reference sample without an FB of virgin PS. As expected, with increasing thickness of the FB, the concentrations of the surrogates decreased due the increasing amount of non-spiked virgin PS. However, the concentration was not constant in the final yogurt cup, but rather a concentration profile within the FB layer. The virgin layer should be less contaminated compared with the spiked core layer of the yogurt cup. Ideally, the concentration in the virgin food contact layer should be virtually zero. However, due to the high temperatures involved in the sheet and yogurt cup manufacturing process as well as the pre-storage of the cup, the FB barrier will most likely be (partially) contaminated with the artificially spiked surrogates. Therefore, there will be a concentration gradient in the FB layer. The concentrations provided in Table 1 are the mean values after exhaustive extraction of the samples. Conclusions about the concentration gradient in the FB layer cannot be drawn from the results.

Styrene is the residual monomer of PS and was not artificially added. Therefore, the concentration of styrene in the five samples is nearly constant (688 ± 29 mg/kg) for all five samples, and there is no concentration gradient in the PS sheet or yogurt cup in the case of styrene. Therefore, the migrated amount of styrene should be comparable for all five samples and independent of the FB thickness. Styrene can, therefore, be considered as an internal standard in the migration kinetic test.

### 2.3. Experimental Migration Kinetics

The migration of the applied surrogates from the co-extruded yogurt cups were determined in a migration kinetic at temperatures of 40 °C and 60 °C over 10 days. For all investigated migrants, kinetic points were determined every 40 min, resulting in a total of 360 kinetic points available for each surrogate and each temperature. In order

to consider the pre-storage time of the unfilled PS cups, the samples were measured approximately seven months after production. During the pre-storage time, the spiked surrogates could migrate from the core layer into the FB layer. The migration curves are shown in Figure 1 (40 °C) and Figure 2 (60 °C), respectively. The migration kinetics at 40 °C were repeated on Samples 3 and 4 after approximately seven additional months of pre-storage (overall 14 months after production, migration kinetics not shown). The amounts of surrogates migrated after storage for 10 days at 40 °C and 60 °C (in µg/cm$^2$) are shown in Tables 2 and 3.

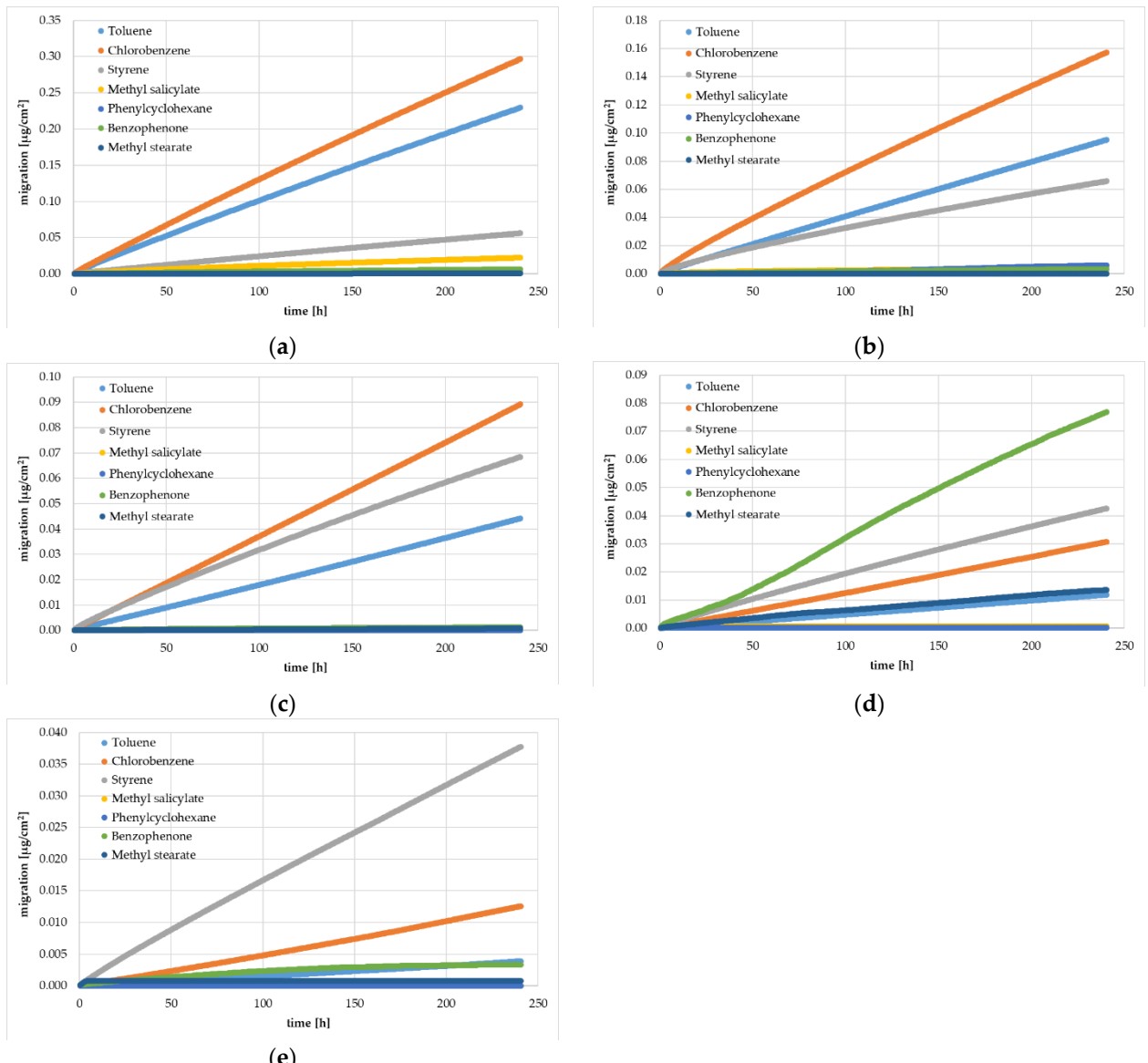

**Figure 1.** Experimentally determined gas phase migration kinetics (360 kinetic points) from HIPS yogurt cups at 40 °C: Sample 0 (**a**), Sample 1 (**b**), Sample 2 (**c**), Sample 3 (**d**), and Sample 4 (**e**). Spiking levels for the different substances are indicated in Table 1.

In Sample 3.1, the surrogate benzophenone shows unexpectedly high results in the amount migrated into the gas phase. This is most likely due to analytical artefacts. This is supported by the fact that Sample 3.2, measured after the additional pre-storage of seven months, shows a typical migration behavior of benzophenone.

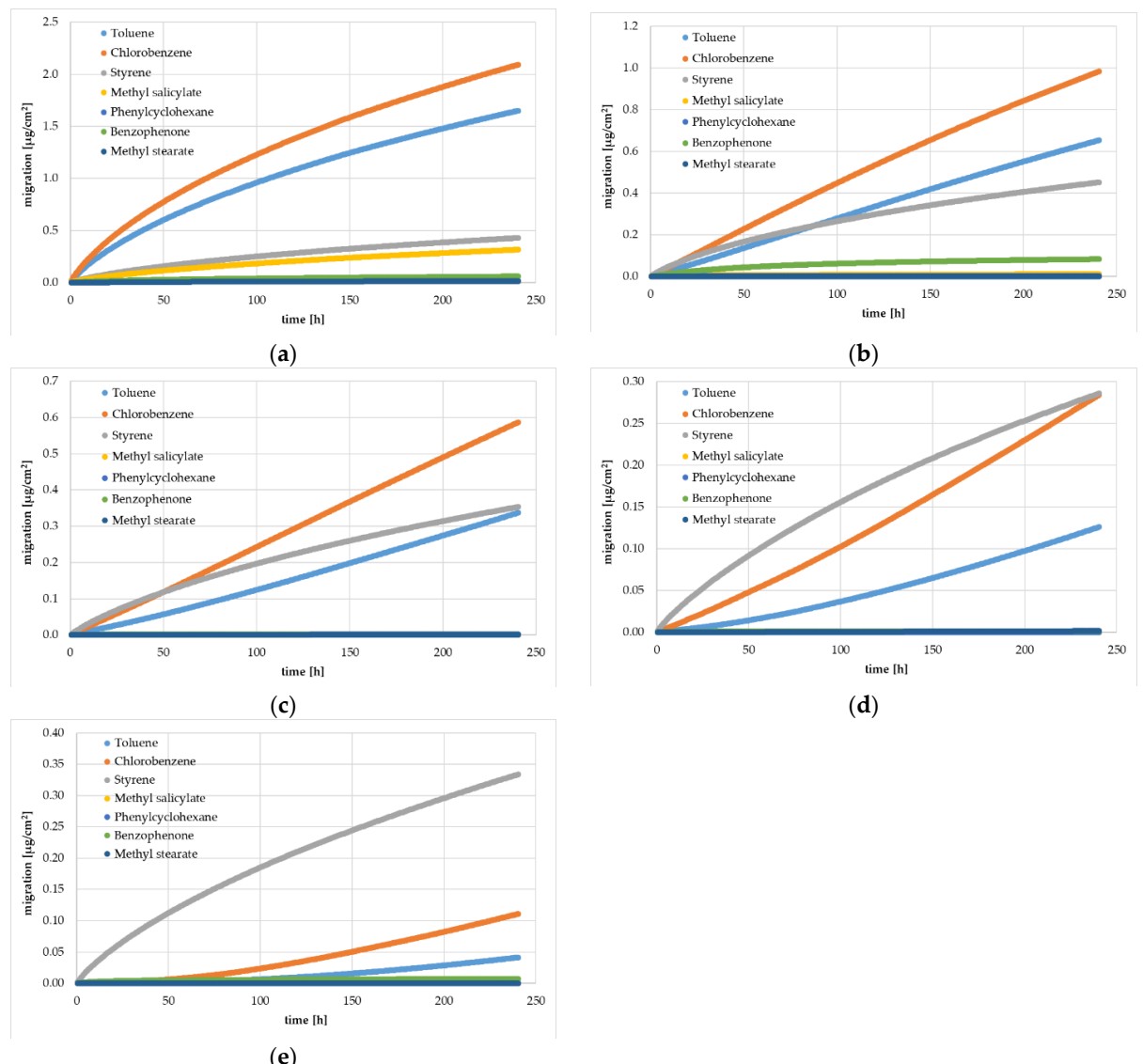

**Figure 2.** Experimentally determined gas phase migration kinetics (360 kinetic points) from HIPS yogurt cups at 60 °C: Sample 0 (**a**), Sample 1 (**b**), Sample 2 (**c**), Sample 3 (**d**), and Sample 4 (**e**). Spiking levels for the different substances are indicated in Table 1.

**Table 2.** Experimentally determined migrated amounts of the surrogates and styrene into the gas phase from yogurt cups after storage for 10 days at 40 °C (pre-storage time of 14 months for Samples 3.2 and 4.2; pre-storage time for all other samples 7 months).

| Sample | Migrated Amount (µg/cm²) | | | | | | |
|---|---|---|---|---|---|---|---|
| | **Toluene** | **Chlorobenzene** | **Styrene** [1] | **Methyl Salicylate** | **Phenyl Cyclohexane** | **Benzophenone** | **Methyl Stearate** |
| Sample 0 | 0.230 | 0.297 | 0.0562 | 0.0224 | 0.00145 | 0.00627 | 0.000709 |
| Sample 1 | 0.0952 | 0.157 | 0.0658 | 0.00366 | 0.00608 | 0.00335 | 0.000168 |
| Sample 2 | 0.0442 | 0.0892 | 0.0685 | <0.0001 [2] | <0.0001 [2] | 0.00119 | 0.000767 |
| Sample 3.1 | 0.0119 | 0.0307 | 0.0426 | 0.000676 | 0.000152 | (0.0769) [3] | 0.0136 |
| Sample 4.1 | 0.00389 | 0.0126 | 0.0377 | <0.0001 [2] | <0.0001 [2] | 0.00336 | 0.000809 |
| Sample 3.2 | 0.0250 | 0.0506 | 0.0414 | 0.000237 | 0.000116 | 0.00113 | <0.0001 [2] |
| Sample 4.2 | 0.00463 | 0.0117 | 0.0166 | <0.0001 [2] | <0.0001 [2] | 0.000160 | <0.0001 [2] |

[1] Residual monomer, not artificially added; [2] Below the experimental detection limit of 0.0001 µg/cm²; [3] Analytical artefacts.

**Table 3.** Experimentally determined migrated amounts of the surrogates and styrene into the gas phase from yogurt cups after storage for 10 days at 60 °C (pre-storage time for all samples seven months).

| Sample | Migrated Amount ($\mu$g/cm$^2$) | | | | | | |
|--------|---------|--------------|-----------|---------------------|---------------------|--------------|-------------------|
|        | Toluene | Chlorobenzene | Styrene [1] | Methyl Salicylate | Phenyl Cyclohexane | Benzophenone | Methyl Stearate |
| Sample 0 | 1.65 | 2.09 | 0.430 | 0.317 | 0.0174 | 0.0604 | 0.0149 |
| Sample 1 | 0.653 | 0.983 | 0.452 | 0.0119 | 0.00260 | 0.0834 | 0.000150 |
| Sample 2 | 0.337 | 0.587 | 0.354 | 0.000196 | <0.0001 [2] | 0.00166 | 0.00189 |
| Sample 3 | 0.126 | 0.284 | 0.286 | <0.0001 [2] | <0.0001 [2] | 0.00103 | 0.00165 |
| Sample 4 | 0.0416 | 0.111 | 0.334 | 0.000324 | <0.0001 [2] | 0.00720 | <0.0001 [2] |

[1] Residual monomer, not artificially added; [2] Below the experimental detection limit of 0.0001 $\mu$g/cm$^2$.

## 3. Discussion

### 3.1. Migration Kinetics

The results of the migration from the artificially spiked yogurt cups were determined by a migration kinetic into the gas phase. As expected, the experimentally determined amount of the surrogates migrated were significantly lower at 40 °C than at 60 °C. All samples showed an increase of the migrated amount over time up to 10 days. For the reference cup without an FB (Sample 0), the kinetic curves were in compliance with Fickian diffusion behavior and with the results from the migration kinetics of the previous study [13]. Fickian diffusion behavior is indicated when the correlation between square root of time and migrated amount shows a (nearly) linear trend, which was the case at a temperature of 60 °C. At the lower temperature of 40 °C, the migration process started a little more slowly, which indicates that the concentration of the surrogates at the surface was slightly lower. This might have been due to evaporation of the surrogates from the hot surface of the sheets during sheet or cup manufacturing. For the reference sample, the diffusion coefficients could be derived for all spiked surrogates from the experimental migration kinetics (see Section 3.2). In addition, the migration kinetics for the residual monomer styrene were similar for all investigated samples with or without an FB layer. This is due to the fact that styrene evenly spread over all layers independent of the thickness of the FB and the spiking level of the surrogates. The diffusion coefficients for styrene could be derived from all investigated samples.

All samples with an FB layer showed a lower migrated amount (except styrene) compared with the reference sample without an FB, which is consistent with the intended role of an FB. However, it can be noticed from Figures 1 and 2 that for the first kinetic point after 40 min, all surrogates were detectable in Samples 1 to 4. A significant delay in the migration curve for passing the FB (the so-called lag time) was not detectable in Samples 1 to 4. This clearly indicates that the FB barrier was contaminated with the surrogates from the core layer. The lack of a lag time also indicates that the surrogates reached the inner surface of the yogurt cups. This contamination of the virgin FB layer can be either from the high temperatures during PS sheet and cup manufacturing or from the migration process during pre-storage of the yogurt cups for seven months, or both. On the other hand, the latter can perhaps be excluded, because Samples 3 and 4 were re-measured after additional seven months pre-storage. Comparing the results of the 7-month pre-storage and the 14-month pre-storage show that the migrated amounts were similar or lower after storage of 14 months, which clearly indicates that the contamination of the FB layer was caused by the high temperatures (in the PS melt) during sheet manufacturing.

As mentioned above, the migration kinetics of reference sample without an FB (Sample 0) followed Fickian diffusion behavior. From the migration kinetics, the diffusion coefficients $D_P$ for the surrogates could be derived. For this purpose, the experimental results provided in Tables 2 and 3 were fitted by use of a diffusion model (see Section 4.4). In addition to the experimental data from this study, the migration kinetics determined by

the same methods from the previous study, Table 1 in [13], were also fitted and correlate with each other. The diffusion coefficients $D_P$ are provided in Table 4. The effect of the temperature on the diffusion coefficient resulted in the activation energy of diffusion $E_A$ and the pre-exponential factor $D_0$ according to the Arrhenius equation. The Arrhenius plots are provided in Figure 3. It should be noted that due to the very low migrated amount of the high-molecular-weight surrogates benzophenone and methyl stearate, the Arrhenius plots show poor correlation. For both substances, activation energies of diffusion could not be derived. Diffusion coefficients $D_P$ and activation energies of diffusion $E_A$ are available only for toluene, chlorobenzene, styrene, and phenyl cyclohexane.

**Table 4.** Diffusion coefficients $D_P$, activation energies of diffusion $E_A$, and pre-exponential factor $D_0$ derived from the migrated amounts.

| Temperature | Diffusion Coefficient $D_P$ (cm²/s) | | | | |
|---|---|---|---|---|---|
| | Toluene | Chlorobenzene | Styrene [1] | Methyl Salicylate | Phenyl Cyclohexane |
| 5 °C [13] | $2.50 \times 10^{-16}$ | $3.80 \times 10^{-16}$ | $3.40 \times 10^{-16}$ | / | $1.87 \times 10^{-19}$ |
| 20 °C [13] | $1.41 \times 10^{-15}$ | $7.67 \times 10^{-15}$ | $3.08 \times 10^{-15}$ | / | $8.30 \times 10^{-19}$ |
| 40° (this study) | $5.85 \times 10^{-14}$ | $8.45 \times 10^{-14}$ | $4.88 \times 10^{-15}$ | $2.23 \times 10^{-16}$ | $9.10 \times 10^{-19}$ |
| 40 °C [13] | $5.15 \times 10^{-14}$ | $1.64 \times 10^{-13}$ | $8.00 \times 10^{-14}$ | / | $5.70 \times 10^{-17}$ |
| 60 °C (this study) | $3.00 \times 10^{-12}$ | $4.20 \times 10^{-12}$ | $2.87 \times 10^{-13}$ | $5.06 \times 10^{-16}$ | $1.32 \times 10^{-16}$ |
| 60 °C [13] | $2.35 \times 10^{-12}$ | $1.65 \times 10^{-11}$ | $2.63 \times 10^{-12}$ | / | $2.19 \times 10^{-15}$ |
| $E_A$ (kJ/mol) | 134.0 | 138.0 | 108.2 | / | 112.7 |
| $D_0$ (cm²/s) | $1.84 \times 10^9$ | $2.42 \times 10^{10}$ | $4.76 \times 10^4$ | / | $1.24 \times 10^2$ |
| $r^2$ | 0.9802 | 0.9679 | 0.8479 | / | 0.7560 |

[1] Residual monomer, not artificially added.

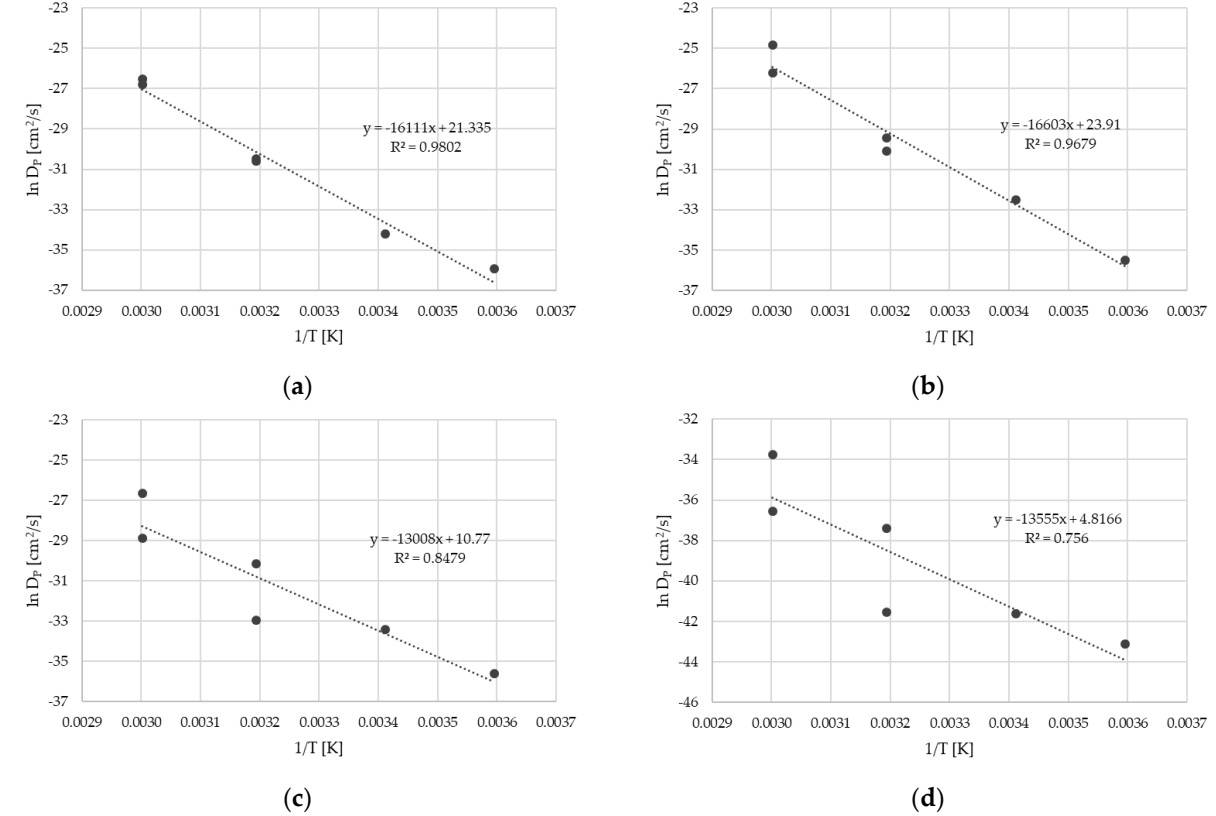

**Figure 3.** Arrhenius plots for toluene (**a**), chlorobenzene (**b**), styrene (**c**), and phenyl cyclohexane (**d**). Diffusion coefficients $D_P$ from Table 4.

### 3.2. Exposure Evaluation

On the basis of the experimentally determined migration data, the exposure to the consumer was evaluated. Two different approaches were used. In the first approach, the 10 μg/kg migration limit was used for the safety evaluation of post-consumer recycled PS in yogurt cups. This migration limit was given as an evaluation criteria in Regulation (EC) 10/2011 [16]. At the end of the shelf life, the concentrations of any contaminants from the first use of the PS yogurt cup (or from misuse), therefore, should be below this migration threshold limit. The corresponding initial concentrations of contaminants in the recyclate-containing PS packaging is, however, not specified in Regulation (EC) 10/2011. The general understanding, however, is that the evaluation of safety should be subject to a worst-case scenario.

The migrations kinetics within this study were performed with extremely high spiking levels of the applied surrogates. The concentration range varied between 264 mg/kg (toluene in Sample 4) and 1389 mg/kg (phenyl cyclohexane in Sample 0). The concentrations of substances from the first use of the package or from misuse are more realistically expected in the lower mg/kg (ppm) range. For example, a concentration of 3 mg/kg was considered as worst-case concentration in the first part of this series [13], which was derived as a pragmatic approach compared with the input concentration for PET [5,18,19]. Most likely, the concentrations of contaminants in post-consumer PS are significantly lower than 3 mg/kg. A discussion of the contamination level of post-consumer PS is provided in the previous study [13]. The applied concentrations for the migration kinetics in this study (Table 1), therefore, are far in excess of the realistic concentration of any contaminants in rPS. Therefore, the migrated amounts should be normalized. This is also important for the comparison of the different FB layer thicknesses, since the concentrations of the surrogates are different in the investigated samples due to different amounts of non-spiked virgin PS. This means that the kinetic results from Tables 2 and 3 are not directly comparable due to their different concentrations in the packages. As a pragmatic approach, the concentrations of the surrogates (from Table 2, 40 °C) were normalized to 30 mg/kg, which is a safety factor of 10 higher than the worst-case concentration of any contaminants assumed in rPS [13]. The safety factor of 10 is recommended for example by [18,24]. In addition to the normalization of the concentration in the package, the migration results are re-calculated from the kinetic data (in μg/cm$^2$) into concentration in the yogurt (in μg/kg) for a typical yogurt cup with 122.55 cm$^2$ inner surface area and a yogurt content of 125 mL. These normalized results (shown in Table 5) can be compared between the different cups and virgin layer thicknesses. As a result, by following this evaluation approach, the migrated amounts for all surrogates are below the threshold limit of 10 μg/kg. It is interesting to note that the reference sample without an FB (Sample 0) is also in compliance with the above-mentioned evaluation criteria, which is in good agreement with the previous study [13].

**Table 5.** Migrated amounts of the surrogates and styrene into the gas phase from yogurt cups after storage for 10 days at 40 °C (pre-storage time for Samples 3.2 and 4.2 is 14 months; pre-storage time for all other samples is 7 months). Spiked amount normalized to 30 mg/kg.

| Sample | Migrated Amount (μg/kg) | | | | | |
|---|---|---|---|---|---|---|
| | Toluene | Chlorobenzene | Methyl Salicylate | Phenyl Cyclohexane | Benzophenone | Methyl Stearate |
| Sample 0 | 7.74 | 9.34 | 0.512 | 0.0306 | 0.135 | 0.0190 |
| Sample 1 | 4.99 | 7.87 | 0.143 | 0.222 | 0.122 | 0.00754 |
| Sample 2 | 2.75 | 5.22 | <0.00452 | <0.00421 | 0.0504 | 0.0401 |
| Sample 3.1 | 0.925 | 2.23 | 0.0379 | 0.00795 | (4.08) [1] | 0.896 |
| Sample 4.1 | 0.433 | 1.31 | <0.00831 | <0.00770 | 0.268 | 0.0790 |
| Sample 3.2 | 1.94 | 3.66 | 0.0133 | 0.00607 | 0.0602 | <0.00658 |
| Sample 4.2 | 0.516 | 1.22 | <0.00831 | <0.00770 | 0.0127 | <0.00977 |

[1] Analytical artefacts.

The second approach for the safety evaluation of rPS behind an FB of virgin PS followed the EFSA criteria for mechanical recycling processes. A maximum migration value of 0.1 µg/kg was derived for a yogurt cup made from 100% PS recyclate [13]. This concentration was calculated for a toddler with 10 kg body weight consuming 250 g yogurt per day (all packed exclusively in PS packaging). The initial concentration in the package was assumed to be 3 mg/kg [13]. A migration limit of 0.1 µg/kg is a very low concentration in food, which will be difficult to detect in migration tests with food simulants and even more difficult to detect in real food. The EFSA approach, therefore, used diffusion modelling for the safety evaluation of recycled polymers, which circumvents the problem of such low migration threshold limits. As mentioned above (Section 2.1), diffusion modelling cannot be applied in the evaluation of co-extruded sheets because the virgin FB layer is contaminated during sheet production. The concentration profile of each of the surrogates in the FB structures is difficult or impossible to determine in experimental tests with food simulants or real yogurt. Therefore, a threshold limit of 0.1 µg/kg is most likely not suitable for experimental migration tests with FB layers with food simulants or real food testing.

The migration testing method into the gas phases applied in this study, however, achieves very low detection limits. The normalized migrated amounts (from Table 2, 40 °C, yogurt cup with 125 mL, 122.55 cm$^2$) are summarized in Table 6. As a result, the migrated amounts for toluene (Samples 0 to 2) and chlorobenzene (all samples) exceed the migration limit of 0.1 µg/kg after storage of the samples for 10 days at 40 °C. The surrogates methyl salicylate, phenyl cyclohexane, benzophenone, and methyl stearate are in compliance with the migration threshold limit of 0.1 µg/kg and a packaging concentration of 3 mg/kg. So, only the small molecules such as toluene and chlorobenzene show migration that is too high. For the interpretation of the results, it is important to note that the migration kinetics were determined at 40 °C. The normal storage conditions for yogurt are refrigerated conditions (e.g., 6 °C), and the storage time is approximately 40 days.

**Table 6.** Migrated amounts of the surrogates and styrene into the gas phase from yogurt cups after storage for 10 days at 40 °C (pre-storage time for Samples 3.2 and 4.2 is 14 months; pre-storage for all other samples is 7 months). Spiked values normalized to 3 mg/kg.

| Sample | Migrated Amount (µg/kg) | | | | | |
| | Toluene | Chlorobenzene | Methyl Salicylate | Phenyl Cyclohexane | Benzophenone | Methyl Stearate |
|---|---|---|---|---|---|---|
| Sample 0 | 0.774 | 0.934 | 0.0512 | 0.00306 | 0.0135 | 0.00190 |
| Sample 1 | 0.499 | 0.787 | 0.0143 | 0.0222 | 0.0122 | 0.000754 |
| Sample 2 | 0.275 | 0.522 | <0.000452 | <0.000421 | 0.00504 | 0.00401 |
| Sample 3 | 0.0925 | 0.223 | 0.00379 | 0.000795 | (0.408) [1] | 0.0896 |
| Sample 4 | 0.0433 | 0.131 | <0.000831 | <0.000770 | 0.0268 | 0.00790 |
| Sample 3.2 | 0.194 | 0.366 | 0.00133 | 0.000607 | 0.00602 | <0.000658 |
| Sample 4.2 | 0.0516 | 0.122 | <0.000831 | <0.000770 | 0.00127 | <0.000977 |

[1] Analytical artefacts.

The activation energies of diffusion $E_A$ (Table 4) can be used to calculate the migration of the surrogates after storage for 40 days at 6 °C. As diffusion modelling requires the concentration profile of a migrant in the package, it is possible to calculate the migrated amount only for a sample without an FB (Sample 0). Consequently, the migrated amount for toluene was 0.080 µg/kg (40 days at 6 °C), starting at a concentration in the package of 3 mg, which is a factor of 9.7 lower than the experimental concentration after storage for 10 days at 40 °C. The same calculations for chlorobenzene resulted in a calculated amount of 0.123 mg/kg, which is factor of 7.6 lower than the experimental value of 0.934 (10 days at 40 °C). Applying both factors to the migrated amounts for toluene and chlorobenzene in Table 6 shows that the migrated amount of toluene is below the 0.1 µg/kg migration limit for all samples. In the case of chlorobenzene, Sample 1 is borderline (0.103 µg/kg), but thicker functional barriers are below the 0.1 µg/kg threshold value. This indicates that

the migration testing conditions 10 days at 40 °C is over-estimative by the factors 9.7 and 7.6, respectively, compared with the migrated amount for 40 days at 6 °C.

## 4. Materials and Methods

### 4.1. Manufacturing of HIPS Sheet and Cups Spiked with Model Compounds

In the first step, the core layer of the co-extruded PS sheets was spiked with model substances (surrogates) during the sheet manufacturing process by first mixing surrogates in virgin PS in melt phase (extrusion) and then using this spiked PS as core layer coextruded with a layer of virgin PS as FB. The following surrogates were applied: toluene, chlorobenzene, methyl salicylate, phenyl cyclohexane, benzophenone, and methyl stearate. In the second step, yogurt cups were manufactured from the spiked sheets. Subsequently, the layer thicknesses were experimentally determined by use of microtome cuts and microscope analysis of the different layers. The results are shown in Table 7.

**Table 7.** Nominal and measured thicknesses of the sheets and yogurt cups.

| Sample | Sheet Thickness | | Barrier Thickness in Cup | | Composition |
|---|---|---|---|---|---|
| | Nominal | Measured | Nominal | Measured | |
| Sample 0 | 800 µm | 790 µm | 0 µm/ 100 µm/ 0 µm | 112 µm | 0%/100%/0% |
| Sample 1 | 720 µm | 163 µm/ 422 µm/ 165 µm | 20 µm/ 50 µm/ 20 µm | 23.1 µm/ 54.0 µm/ 21.3 µm | 22%/56%/22% |
| Sample 2 | 880 µm | 242 µm/ 405 µm/ 231 µm | 30 µm/ 50 µm/ 30 µm | 31.5 µm/ 58.7 µm/ 36.0 µm | 27%/46%/27% |
| Sample 3 | 1040 µm | 320 µm/ 431 µm/ 300 µm | 40 µm/ 50 µm/ 40 µm | 47.6 µm/ 67.1 µm/ 47.4 µm | 30%/40%/30% |
| Sample 4 | 1200 µm | 405 µm/ 406 µm/ 382 µm | 50 µm/ 50 µm/ 50 µm | 53.6 µm/ 55.9 µm/ 55.0 µm | 33%/33%/33% |

The yogurt cup samples were produced on 28 September 2021. The migration testing was performed in May 2022. The pre-storage time was, therefore, approximately 7 months. The migrations tests on Sample 3 and 4 were repeated in December, which corresponds to 14 months pre-use stock.

### 4.2. Quantification of Spiking Levels in the PS Cups

The concentrations of the surrogates were determined in the PS yogurt cups after solvent extraction. For that purpose, 1.0 g of sample material was immersed with 10 mL acetone and stored at 60 °C for 3 days. Subsequently, the extracts were decanted from the polymer and analyzed by gas chromatography with flame ionization detection (GC-FID): Column: MX 1–30 m, i.d.: 0.25 mm, film thickness: 0.25 µm, temperature program: 50 °C (2 min), followed by heating at 10 °C/min to 340 °C (10 min), pre-pressure: 50 kPa hydrogen, and split: 10 mL/min. Calibration was achieved by external calibration using standard solution of the neat surrogates in acetone. *tert*-Butylhydroxyanisole (BHA, CAS No. 8003-24-5) and Tinuvin 234 (CAS No. 70321-86-7) were used as internal standards. The concentrations of the model compounds in the investigated PS cups are summarized in Table 1.

*4.3. Migration Kinetics into the Gas Phase*

The migrated amounts of the surrogates into the gas phase were determined into the gas phase using an automated method. The spiked PS cups were placed in a migration cell in such a way that only the inner surface of the yogurt cup, which is typically in contact with the yogurt, is relevant for the migration. The migration cell was placed in a climate chamber (40 °C and 60 °C). The surrogates migrating from the PS cups were purged out of the migration cell by a helium stream of 20 mL/min. Subsequently, the surrogates were trapped (Carbopack B 40 mm, Supelco) at a trap temperature of −46 °C. The loaded trap was completely desorbed and transferred directly to the connected gas chromatograph (GC) by heating it to 340 °C within approximately 10 s. The surrogates were separated and quantified during the GC run. Calibration was achieved by injection of undiluted standard solutions of the migrants into the migration cell. Gas chromatograph: Column: Rxi624, length: 30 m, inner diameter: 0.32 mm, film thickness: 1.8 μm, temperature program: 40 °C (2 min), followed by heating at 20 °C/min to 280 °C (7 min), pressure: 70 kPa helium, and detector temperature: 280 °C.

*4.4. Diffusion Modelling*

Diffusion coefficients were fitted by use of the AKTS SML software version 4.54 (AKTS AG, Siders, Switzerland). The partitioning coefficient was assumed to be $K_{P,F} = 1$, which indicates high solubility for the surrogates in food. In addition, a density of HIPS of 1.04 g/cm$^3$ was used for the fitting. The wall thickness in the calculations was 300 μm, which is the worst-case scenario for most of the packaging. It should be noted that due to the low diffusivity of HIPS, the partition coefficient as well as the wall thickness had no influence on the (calculated) result.

**5. Conclusions**

In this study, co-extruded yogurt cups with a core layer of post-consumer PS behind an FB of virgin PS were investigated with respect to migration of potential contaminants from the first use. Artificially spiked surrogates into the core layer were used to simulate contaminants. Due to the high processing temperatures used during co-extrusion, the virgin PS FB layer was partly contaminated by the surrogates. The lack of any lag time in the migration process indicated that the surrogates had already reached the inner surface of the yogurt cups, even when a layer thickness of 55 μm was applied. Since the concentration profile of the surrogates was not available, diffusion modelling as typically applied by EFSA could not be used for the performance evaluation of the different FB thicknesses. Experimental migration testing of the surrogates, therefore, was required. The automated migration kinetic testing method used was shown to be sensitive and suitable for the evaluation of the FB performance of co-extruded yogurt cups.

Due to the lack of any guidance or evaluation criteria by national competent authorities or by EFSA, the FB evaluation was conducted in accordance with the evaluation of mechanical recycling processes. In the first approach, the FBs were evaluated with a worst-case concentration (3 mg/kg) of surrogate in the package and an additional safety factor of 10, resulting in a concentration in the package of 30 mg/kg. The migration threshold was derived from Regulation (EC) 10/2011 to be 10 μg/kg after storage for 10 days at 40 °C. According to this approach, all investigated yogurt cups were in compliance with these criteria, including the cup without a functional barrier. In the second approach, the migration was evaluated by a scenario which is in accordance with the EFSA criteria for mechanical recycling processes for PET. The worst-case packaging concentration was again assumed to be 3 mg/kg, and the maximum migration limit was assumed to be 0.1 μg/kg. Within this evaluation scenario, storage for 10 days at 40 °C exceeded the migration limit for the volatile surrogates toluene (FB layer thicknesses < 36.0 μm) and chlorobenzene (up to 55.0 μm). The predicted migrated amounts after storage for 40 days at 6 °C, however, met the very low 0.1 μg/kg migration limit for FB barrier thicknesses of >21.3 μm. This compliance test included a pre-storage time of the empty yogurt cups of 14 months.

As an overall conclusion, migration kinetics into the gas phase with hundreds of kinetic points is a very useful tool for the FB evaluation. The gas phase contact minimizes the interactions with food simulants, some of which swell styrene polymers and might destroy the FB effect. The migration kinetics provide useful insight in the diffusion behavior of potential contaminants in PS. In cases where Fickian laws of diffusion are valid, the diffusion coefficients $D_P$ as well as the activation energies of diffusion $E_A$ are available, which can be used for the compliance evaluation of PS cups with recyclate content. Another conclusion is that for the evaluation of PS recyclates, only the low-molecular-weight solvents play a role in migration, which has an influence on the monitoring program necessary for the introduction of novel technologies notifications. As an overall conclusion, on the basis of the conditions, data, and evaluation presented in this study, the use of post-consumer recycled PS behind an FB can be considered as safe.

**Funding:** This research received no external funding.

**Institutional Review Board Statement:** Not applicable.

**Informed Consent Statement:** Not applicable.

**Data Availability Statement:** Not applicable.

**Acknowledgments:** This study was financially supported by Styrenics Circular Solutions (SCS), Belgium. Thanks are due to Silvia Demiani, Johann Ewender, and Anita Gruner for experimental contributions to this work. Special thanks are due to Claudio Bilotti (Versalis), Frank Eisenträger (Ineos Styrolution), Ken Huestebeck and Jens Kathmann (both Styrenics Circular Solutions), and Mark Pemberton (Systox) for fruitful discussions and proof-reading of the manuscript.

**Conflicts of Interest:** The author declares no conflict of interest.

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
