# Peer review of "Recycling of Post-Consumer Polystyrene Packaging Waste into New Food Packaging Applications—Part 2: Co-Extruded Functional Barriers"

_recycling, doi:10.3390/recycling8020039_

Round 1

Reviewer 1 Report

This is an excellent work that investigates the decontamination of polystyrene packaging during the recycling process. It is generally well written and well discussed.

However, it is necessary for the author to include the main results in the Abstract. In addition, the figures (graphs) are in gray color, I suggest you make an edition to Black color to improve the quality of the figures.

Author Response

This is an excellent work that investigates the decontamination of polystyrene packaging during the recycling process. It is generally well written and well discussed.

FW: Thanks for your valuable review.

However, it is necessary for the author to include the main results in the Abstract.

FW: According to the Journals Guidelines the abstract is restricted to about 200 words in maximum. In the submitted version the abstract consists of 188 words. So there is not much space to extent the abstract. In my point of view the main findingsare presented in the abstract: The functional barrier is contaminated during coextrusion, but the yogurt application can be considered as safe. I revised the abstract a little bit to make the key findings more visible to the reader.

In addition, the figures (graphs) are in gray color, I suggest you make an edition to Black color to improve the quality of the figures.

FW: Thanks for this comment. I changed the colours to black.

Reviewer 2 Report

The reviewed paper presents a continuation of the research previously published in the journal "Recycling". The whole has been prepared in a similar way and can be published after linguistic and editorial control.

Author Response

The reviewed paper presents a continuation of the research previously published in the journal "Recycling". The whole has been prepared in a similar way and can be published after linguistic and editorial control.

FW: Thanks for your valuable review.

Author Response

The paper concerns the assessment of the suitability for packaging applications of post-consumer polystyrene (PS) recyclates behind a functional barrier (with different thickness) of virgin PS polymer. The topic of the research is of great interest and the author performed an extensive and detailed plan of experimental tests. In the referee’s opinion the paper deserves to be published in “Recycling” journal, but beforehand the author should clarify one important issue.

FW: Thanks for your valuable review.

At page 14, lines 493-496, he concludes that “...migration kinetics into the gas phase with hundreds of kinetic points is a very useful tool for the FB evaluation. The gas phase contact minimizes the interactions with food simulants some of which swell styrene polymers and might destroy the FB effect.” The referee’s concern is just about the suitability of this technique used for the migration measurements. In particular, according to the Regulation EU 2016/1416, the migration tests should be conducted using specific simulants at defined contact conditions, depending on the kind of food to be packed with the material under testing. In this study, the author refers to PS cups for yogurt, that is an acid food with a high liquid content and, as consequence, its extractive ability should be different (most probably higher) than the one of the gas helium, used in this study for the migration tests. The referee strongly suggests to clarify this aspect.

FW: Indeed Regulation 10/2011 and its amendments (e.g. 2016/1416) requires that migration contact be simulated with food simulants. The food simulant is 50% ethanol and the contact conditions for yogurt is 10 d at 40 °C. PS is very sensitive towards solvents like ethanol. In a previous study we conducted migration kinetics into 50% ethanol as well as other simulants like iso-octane and 95% ethanol. The weight increase of the HIPS at 40 °C with 50% ethanol was 1.2% (Guazzotti et al, Migration Testing of GPPS and HIPS Polymers: Swelling Effect Caused by Food Simulants Compared to Real Foods, Molecules 2022, 27, 823, reference 23 in the current manuscript). This increase is most probably at the surface, which means that the first couple of µm is penetrated during the simulant contact. This increases the diffusion and destroys the functional barrier effect. In the same publication, the weight increase in contact with milk and cream (also acid food) was determined to 0.1% and <0.1% So acid milk products does not swell the polymer surface. Therefore food simulants like 50% ethanol are not suitable for functional barrier performance testing. Direct testing into milk products is difficult due to spoilage at 40 °C. Without swelling effect in contact with acid milk products, potential migrants follow Fickian laws of diffusion. This is what had been determined with the migration kinetics into gas phase, best seen in the kinetic with styrene. I integrated this discussion into section "study design" of the manuscript.

On the other hand, the same concern was already expressed in the case of the author’s previous paper “Recycling of post-consumer polystyrene packaging waste into new food packaging applications – Part 1: Direct food contact”. In the following, the reviewer’s comment on the first paper: “The over-estimated factors of the model migration predicted values compared to the experimental ones are very high for some surrogates, specifically at the lower temperatures investigated: maybe it would more suitable to measure the specific migration values using the experimental procedure established within the Regulation EU 2016/1416.”

FW: As explained above, migration into food simulants is not suitable for testing of migration . By doing that, the over-estimative factors is indeed expected to be lower. But this is due to swelling of PS with 50% ethanol, but again, acid milk products do not swell the HIPS matrix. It should be noted here, that the most important value is the migration into real food and NOT into simulants like 50% ethanol at elevated temperatures. Temperature increase to 40 °C is applied to speed up the experimental migration tests. The over-estimative factors are not discussed in this manuscript. Therefore no changes had been made in the manuscript according to this point.

Round 2

Reviewer 3 Report

The author adequately addressed the reviewer' comments, so the paper can be published in the present form.